



**Low sensitivity of gross primary production to elevated CO2 in a mature Eucalypt woodland**
**Authors:** Jinyan Yang[1], Belinda E. Medlyn[1], Martin G. De Kauwe[2,3], Remko A. Duursma[1], Mingkai Jiang[1],
Dushan Kumarathunge[1], Kristine Y. Crous[1], Teresa E. Gimeno[4,5], Agnieszka Wujeska-Klause[1], David S.
Ellsworth[1]
**Affiliation**: [1] Hawkesbury Institute for the Environment, Western Sydney University, Penrith, NSW, Australia
[2] ARC Centre of Excellence for Climate Extremes, Sydney, NSW 2052, Australia
[3] Climate Change Research Centre, University of New South Wales, Sydney, NSW 2052, Australia
[4] Basque Centre for Climate Change, Scientific Campus of the University of the Basque Country, Leioa, Spain
[5] IKERBASQUE, Basque Foundation for Science, 48008, Bilbao, Spain
Correspondence to: Jinyan Yang (jinyan.yang@westernsydney.edu.au)
**For submission to: Biogeosciences Discussions**
No of words in abstract: 269
No of words in main text: 6705
No of Figures: 8
No of Tables: 1



**Abstract**

The response of mature forest ecosystems to rising atmospheric carbon dioxide concentration ($C_a$) is a major uncertainty in projecting the future trajectory of the Earth's climate. Although leaf-level net photosynthesis is typically stimulated by exposure to elevated $C_a$ (e$C_a$), it is unclear how this stimulation translates into carbon cycle responses at whole-ecosystem scale. Here we estimate a key component of the carbon cycle, the gross primary productivity (GPP), of a mature native Eucalypt forest exposed to Free Air $CO_2$ Enrichment (the EucFACE experiment). In this experiment, light-saturated leaf photosynthesis increased by 19% in response to a 38% increase in $C_a$. We used the process-based forest canopy model, MAESPA, to upscale these leaf-level measurements of photosynthesis with canopy structure to estimate Gross Primary Production (GPP) and its response to e$C_a$. We assessed the direct impact of e$C_a$, as well as the indirect effect of photosynthetic acclimation to e$C_a$ and variability among treatment plots via different model scenarios.

At the canopy scale, MAESPA estimated a GPP of 1574 g C m$^{-2}$ yr$^{-1}$ under ambient conditions across four years and a direct increase in GPP of +11% in response to e$C_a$. The smaller canopy-scale response simulated by the model, as compared to the leaf-level response, could be attributed to the prevalence of RuBP-regeneration limitation of leaf photosynthesis within the canopy. Photosynthetic acclimation reduced this estimated response to 10%. Considering variability in leaf area index across plots, we estimated a mean GPP response to e$C_a$ of 6% with a 95% CI of (-2%, 14%). These findings highlight that the GPP response of mature forests to e$C_a$ is likely to be considerably lower than the response of light-saturated leaf photosynthesis. Our results provide an important context for interpreting e$C_a$ responses of other components of the ecosystem carbon cycle.



## 1. Introduction

Forests represent the largest long-term terrestrial carbon storage (Bonan, 2008; Pan et al., 2011). Atmospheric carbon dioxide concentration ($C_a$) has increased significantly since the beginning of the industrial era (Joos and Spahni, 2008), but the increase would have been considerably larger without forest carbon sequestration, which is estimated to have offset 25-33% of recent anthropogenic $CO_2$ emissions (Le Quéré et al. 2017). $C_a$ is projected to continue to increase by 1-5 µmol mol$^{-1}$ per year into the future (IPCC, 2014), but the rate of this rise depends on the magnitude of the forest feedback on $C_a$. At the leaf scale, the direct physiological effects of rising $C_a$ are well understood: elevated $C_a$ (e$C_a$) stimulates plant photosynthesis (Kimball et al. 1993; Ellsworth et al. 2012) and reduces stomatal conductance (Morison, 1985, Saxe et al. 1998), which together increase leaf water-use efficiency (De Kauwe et al. 2014).  These physiological responses at leaf scale could potentially increase ecosystem carbon uptake and hence the amount of carbon stored in the ecosystem, which at the global scale significantly mitigates the rise in $C_a$. However, projecting the response of the terrestrial carbon sink to future increases in $C_a$ is a major uncertainty in models (Friedlingstein et al. 2014), highlighting an urgent need to make greater use of data from manipulative experiments at leaf scale to inform terrestrial biosphere models (Medlyn et al., 2015).

Our understanding of ecosystem responses to e$C_a$ relies on both experiments and observations. However, results from different types of studies show some important areas of disagreement. At the global scale, satellite data provide evidence of a strong greening trend over the last 20 years, indicating an increase in leaf area and/or above-ground biomass, which has been attributed to the gradual increase in $CO_2$ (Donohue et al., 2009; Donohue et al., 2013; Yang et al., 2016; Zhu et al., 2016). A positive response of carbon uptake/greenness is also found in manipulative e$C_a$ open-top chamber experiments with young trees (Eamus and Jarvis, 1989; Curtis and Wang 1998; Saxe et al. 1998; Medlyn et al., 1999) and ecosystem-scale FACE experiments in young, aggrading forest stands (Ainsworth and Long, 2005; Norby et al., 2005; , Ellsworth et al. 2012; Walker et al. 2019). In contrast, individual-tree experiments with mature trees (>30 years old) have found relatively small responses of tree growth to e$C_a$ despite an apparent increase in leaf photosynthesis (Dawes et al., 2011; Sigurdsson et al., 2013; Klein et al., 2016). Also, tree-ring studies indicate an apparent lack of stimulation of vegetation growth in mature forests over the last century (Peñuelas et al. 2011; Silva and Anand, 2013; van der Sleen et al. 2014). These studies raise important questions about how mature ecosystems will respond to e$C_a$.

The Eucalyptus FACE experiment (EucFACE; Australia) is the first replicated, ecosystem-scale experiment where a mature native forest has been experimentally subjected to e$C_a$ and provides a valuable case study to assess the response of a mature forest response to e$C_a$ under field conditions (Ellsworth et al. 2017). Results from the first five years (2013-2018) of leaf gas exchange measurements showed a consistent stimulation of leaf-level light-saturated net photosynthesis ($A$) of 19% (Ellsworth et al., 2017; Wujeska-Klause et al., 2019). Nevertheless, the increase in $A$ did not lead to a detectable change in above-ground growth (Ellsworth et al., 2017). These experimental results are consistent with empirical evidence arising from tree-ring studies (Peñuelas et al. 2011; Silva and Anand, 2013; van der Sleen et al. 2014) and also with experimental evidence from individual mature trees (Körner et al., 2005; Dawes et al., 2011; Klein et al., 2016).



As a first step towards reconciling the e$C_a$ responses of leaf photosynthesis and above-ground growth in this
experiment, here we quantify how the whole canopy carbon uptake, or gross primary productivity (GPP) was
increased under e$C_a$. The response of GPP is important because it provides an upper bound on the potential
response of other components of ecosystem carbon balance, such as above-ground growth. It needs to be
quantified explicitly because the response of GPP to e$C_a$ may be quite different to that of leaf net
photosynthesis. The leaf-level response of photosynthesis to e$C_a$ is usually measured on sunlit leaves under
saturating light (Ainsworth and Rogers, 2007). As a result, these leaf-level e$C_a$ responses largely reflect the
responses of the photosynthesis rate when limited by maximum Rubisco activity ($V_{cmax}$). However, depending
on the canopy architecture and ambient light condition, the canopy could have many shaded leaves, which
would mean that the emergent rate of photosynthesis could actually be limited by RuBP regeneration ($J$). RuBP-
regeneration limited photosynthesis has a smaller response to e$C_a$ than Rubisco-limited photosynthesis
(Ainsworth and Rogers, 2007), resulting in a smaller response of GPP than leaf photosynthesis under saturating
light.
The transition from RuBP-regeneration to Rubisco-limited photosynthesis of the canopy is determined by the
ratio of the maximum capacities for RuBP-regeneration and Rubisco activity, $J_{max}$ and $V_{cmax}$ (Friend, 2001;
Zaehle et al. 2014; Rogers et al., 2017). Wullschleger (1993) reported a $J_{max}$:$V_{cmax}$ ratio of 2, which has been
widely adopted in models (e.g., Wang et al., 1998; Luo et al., 2001; Rogers et al., 2017). However, recent
studies have suggested a lower $J_{max}$:$V_{cmax}$ ratio for many forest ecosystems (Kattge and Knorr, 2007; Ellsworth
et al., 2012; Kumarathunge et al., 2018). A lower $J_{max}$:$V_{cmax}$ ratio results in more frequent RuBP-regeneration
limitation of photosynthesis, which reduces the response of GPP to e$C_a$.
It is difficult to directly measure the e$C_a$ effect on GPP. In some previous e$C_a$ experiments, GPP has been
estimated by scaling up from leaf-level measurements using a canopy model. Wang et al (1998) and Luo et al
(2001) both used the tree array model, MAESPA, which can simulate the radiative transfer within and between
tree crowns and can be parameterised to describe the spatial locations and sizes of trees in e$C_a$ experiments. In
these previous applications of MAESPA, the direct response of GPP to e$C_a$ was consistently half of that
observed at the leaf level because of a large contribution of RuBP-regeneration limited photosynthesis to GPP
(Wang et al., 1998; Luo et al., 2001). However, the direct effect of e$C_a$ on photosynthesis was modified by two
major indirect effects. When LAI increased under e$C_a$, the additional leaf area amplified the GPP response by up
to 60%. The other factor is the downregulation of photosynthesis under e$C_a$, or photosynthetic acclimation
(Long et al., 2004; Ainsworth and Rogers, 2007; Rogers, et al., 2017). Under long-term exposure to e$C_a$, some
plants have been observed to reduce nitrogen allocation to Rubisco, which results in a decrease of
photosynthetic capacity (Gunderson and Wullschleger, 1993). The average decrease of $V_{cmax}$ among plants in
FACE experiments was found to be 13% for all species and 6% for trees (Ainsworth and Long, 2005). Both
Wang et al. (1998) and Luo et al. (2001) tested the impact of photosynthetic acclimation and showed a moderate
reduction of canopy GPP (5-6%) due to photosynthetic acclimation (10-20%) at the studied experiments.
Following Wang et al. (1998) and Luo et al. (2001), we used MAESPA (Duursma and Medlyn, 2012) to
estimate canopy GPP at EucFACE in ambient and elevated $C_a$ treatments. The model has previously been
evaluated with leaf- and whole-tree- scale measurements from EucFACE (Yang et al., in review). Here, we first
parameterised the model with physiological, structural and meteorological data measured during the experiment.





Then, we quantified the response of canopy GPP to e$C_a$ and partitioned this response into the direct stimulation
of GPP and the indirect effects of photosynthetic acclimation and variation of LAI. The overall goal of this
study was to estimate the magnitude of the response of forest canopy GPP to e$C_a$ in order to provide a baseline
against which to compare changes in other components of the ecosystem carbon balance.
**2.   Methods**
**2.1 Site**
The EucFACE experiment (technical details in Gimeno et al., 2016) is located in western Sydney, Australia
(33.617S, 150.741E). It consists of six circular plots, each of which has a diameter of 25 m, enclosing 15-25
mature forest trees (referred to as 'rings' hereafter). The rings are divided into two groups: control (with ambient
$C_a$; 390-400 μmol mol$^{-1}$ during the study period) and experimental (e$C_a$; +150 μmol mol$^{-1}$). The tree canopy is
dominated by *Eucalyptus tereticornis* Sm. which are ~20 m in height and have a basal area of ~24 m$^2$ ha$^{-1}$. The
site receives a mean annual precipitation of 800 mm yr$^{-1}$, a mean annual photosynthetically active radiation
(PAR) of 2600 MJ m$^{-2}$ yr$^{-1}$, and a mean annual temperature of 17 ℃.
**2.2 Model**
The MAESPA model is a process-based tree-array model (Wang and Jarvis, 1990) that calculates canopy carbon
and water exchange (https://bitbucket.org/remkoduursma/maespa/src/Yang_et_al_2019/). At each 30-minute
timestep, the model simulates the radiative transfer, photosynthesis, and transpiration of individual trees
mechanistically. Soil moisture balance can be calculated dynamically, but here we chose to improve accuracy by
using soil moisture as an input to the model (Duursma and Medlyn, 2012).
The model represents the tree canopy as an array of tree crowns. The location and dimensions of each crown are
specified based on-site measurements (see 2.3.2 Canopy structure, below). Calculations of carbon and water
fluxes are made for each tree crown, which is divided into six layers. Here it was assumed that crowns are
represented by an ellipsoidal shape and that leaf area is uniformly distributed across layers within the tree
crown. The leaf angles were assumed to follow a spherical distribution to ensure consistency with the method
used to estimate leaf area index (LAI) in Duursma et al. (2016). Within each layer, the model evaluates the
radiation transfer and leaf gas exchange at 12 grid points such that each crown is represented by a total of 72
grid points. The radiation intercepted at each grid point is calculated for direct and diffuse components by
considering shading from the upper crown and surrounding trees and solar angle (zenith and azimuth), and light
source (diffuse or direct). Penetration by direct radiation to each grid point is used to estimate the sunlit and
shaded leaf area at each grid point. The radiation intercepted by the fraction of sunlit and shade foliage is then
used to calculate the leaf gas exchange.
The gas exchange sub-model combines the leaf photosynthesis model of Farquhar et al. (1980) with the stomatal
optimisation model, following Medlyn et al. (2011). Stomatal conductance is modelled as:
$$g_s = 1.6 \cdot \left(1 + \frac{g_1}{\sqrt{D}}\right) \cdot \frac{A_{net}}{C_a} \tag{1}$$
where $g_s$ is the stomatal conductance to water vapour (mol m$^{-2}$ s$^{-1}$); $g_1$ is a parameter that represents the $g_s$
sensitivity to photosynthesis (kPa$^{0.5}$; see definition in Medlyn et al., (2011)); $A_{net}$ is the net CO$_2$ assimilation rate



($\mu$mol m$^{-2}$ s$^{-1}$); $C_a$ is the atmospheric $CO_2$ concentration ($\mu$mol mol$^{-1}$). The factor 1.6 converts the conductance of
$CO_2$ to that of $H_2O$.
The impact of soil moisture on $g_s$ is represented through an empirical function that links soil water availability
to $g_1$ following (Drake et al., 2017):
$$g_1 = g_{1.max} \left(\frac{\theta - \theta_{min}}{\theta_{max} - \theta_{min}}\right)^q \tag{2}$$
where the $g_{1.max}$ is the maximum $g_1$ value; $\theta$ is volumetric soil water content (%); $\theta_{max}$ and $\theta_{min}$ are the upper and
lower limit within which $\theta$ has impact on $g_1$; $q$ describes the non-linearity of the curve. The equations to
calculate $A_{net}$ are in Supplementary (Text S1, Eqns. S1 – S6).
Following Yang et al. (2019), MAESPA considers a non-stomatal limitation to biochemical parameters $J_{max}$ and
$V_{cmax}$ at high $D$:
$$V_{max} = V_{max.t}(1 - c_D \cdot D) \tag{3}$$
where $V_{max.t}$ is the $J_{max}$ or $V_{cmax}$ at given leaf temperature, and $c_D$ is a fitted parameter (Table 1). This relationship
is empirical and fitted to data collected in EucFACE. Incorporating this relationship was shown to improve the
predicted photosynthesis by the leaf gas exchange model (Yang et al., 2019).
Combining Eqns. 1- 3 and S1 – S6 yields the $g_s$ and $A_{net}$ of each grid point, which is then multiplied by leaf area
at each grid point and summed to give whole-tree photosynthesis. Photosynthesis of individual trees is then
summed to give whole-canopy photosynthesis.
**2.3 Model Parameterisation**
*2.3.1 Meteorological forcing*
The model is driven by *in situ* PAR, wind speed, air temperature, vapour pressure deficit ($D$), and soil moisture
measurements from 2013 to 2016 (Figures 1 and 2). The PAR, air temperature, and relative humidity were
measured every five minutes in each ring and then were gap-filled by linear interpolation and aggregated to 30
minute-mean time slices across all six rings (Figure 1). Each ring has a set of PAR (LI-190, Li-cor, Lincoln, NE,
U.S.), wind speed (Wincap Ultrasonic WMT700 Vaisala, Vantaa, Finland), humidity, and temperature sensors
(HUMICAP ® HMP 155 Vaisala, Vantaa, Finland) at the centre of the ring above the canopy at 23.5 m. $D$ was
calculated from temperature and humidity measurements.
Two levels of $C_a$ were used in the model according to the measured $C_a$ (LI-840, Li-cor, Lincoln, NE, U.S.). The
ambient $C_a$ was gap-filled (in total <10 days during four years gaps due to power outage) and aggregated to 30
minute-mean time slices from the five-minute measurements across the three ambient rings (rings 2, 3, and 6).
The e$C_a$ was processed in the same way but using data from the experimental rings (rings 1, 4, and 5).
The volumetric soil water content ($\theta$) was used as an estimate of plant water availability and was taken every 20
days using neutron measurements at 25 cm intervals (503DR Hydroprobe, Instroteck, NC, U.S.) and averaged to
the top 150 cm (Figure 2). There were two probes in each ring and the average of these probes was used to
represent the ring average for each measurement date. $\theta$ was updated on the days of measurements and thus not
gap-filled.





*2.3.2 Canopy structure*
Trees in MAESPA were represented by their actual location, height, and crown size to mimic the realistic
effects of shading. Tree location, crown height, crown base and stem diameter were measured in January 2013
at the start of the experiment. For each ring, a time-series of LAI was obtained based on measurements of
above- and below- canopy PAR (Duursma et al. 2016). This LAI represents plant area index, which includes the
woody component as well as leaves and does not account for clumping. In order to retrieve the actual LAI, we
assumed a constant branch and stem cover (0.8 $m^2$ $m^{-2}$) based on the lowest LAI during November 2013 when
the canopy shed almost all leaves. The LAI used in this study was thus the plant area index estimates from
Duursma et al. (2016), less 0.8 $m^2$ $m^{-2}$ (Figure 2a). Since LAI is the only parameter beside soil moisture that
differed by ring, canopy structure (i.e., the LAI and its distribution) was the major driver of inter-ring
variability.
The total leaf area ($m^2$) of each ring was calculated as the product of LAI and ground area of each plot (491 $m^2$).
This total leaf area (LA) was then assigned to each tree based on an allometric relationship between the total leaf
area ($m^2$) and diameter at breast height (DBH; m). The allometric relationship was derived from data in the
BAAD database (Falster et al., 2015) for *Eucalyptus* trees grown in natural conditions with DBH <1 m to match
the characteristics of EucFACE. In total, this database yielded a total of 66 observations with which to estimate
the relationship between LA and DBH:
$$L_{allom} = a \cdot DBH^b \tag{4}$$
where $L_{allom}$ is the theoretical leaf area based on allometric relationship to DBH. The values obtained via fitting
for $a$ and $b$ were 492.6 and 1.8 respectively, with a root mean square error of 14.4 ($m^2$). This relationship was
used to assign the total LA of each ring to each tree in the following steps: (i) the $L_{allom}$ for each tree was
calculated based on DBH; (ii) the $L_{allom}$ was summed to obtain a total LA for each ring; and (iii) the fractional
contribution of each tree to the ring total LA was calculated. The total LA based on LAI was then assigned to
each tree based on this fraction.
The crown radius was calculated with a linear function with DBH based on measurements made in August
2016. The data consisted of DBH and crown radius (one on North-South axis and one on East-west axis) of four
trees in each ring. The crown radius measurements were averaged by tree and used to fit a linear model with
DBH. The estimated slope and intercept of the relationship are 0.095 (m $cm^{-1}$) and 0.765 (m), respectively.
MAESPA also considered the shading from surrounding trees outside the rings. However, no measurements of
locations or diameters were available for the trees surrounding the rings. Therefore, a total of 80 surrounding
trees were arbitrarily assumed to form two uniform and circular layers around each ring. They were assigned the
mean height, mean crown radius, and mean leaf area estimated from all trees in EucFACE. Except for shading,
the surrounding trees have no impact on the trees within the rings. Ring 1 is shown in Figure S1 as an example
of the representation of canopy structure in MAESPA.
*2.3.3 Physiology*
The physiological parameters were estimated from field gas exchange measurements as described below. The
data were collected with portable photosynthesis systems (Li-6400, Li-Cor, Inc., USA). The only parameter



found to differ between ambient and elevated $C_a$ rings was $V_{cmax.25}$ ($V_{cmax}$ at 25 ºC; Ellsworth et al., in prep.).
Hence, all other parameters (e.g., the temperature responses of photosynthesis and respiration) were estimated
by combining all data across $CO_2$ treatments. Fitted parameter values are given in Table 1.
A set of temperature-controlled photosynthesis-$CO_2$ response ($A$-$C_i$) curves was measured at different leaf
temperatures (20-40 ºC) under saturating light in February 2016. The dataset was used to quantify the
temperature dependences of $J_{max}$ and $V_{cmax}$ by fitting a peaked Arrhenius function (Eqn. S5) to the
measurements. We assumed that these temperature response functions applied throughout the period of the
study.
Light- and temperature-controlled $A$-$C_i$ curves were also measured in the morning for ten field campaigns
during 2013 to 2016. All $A$-$C_i$ curves were started at the growth $C_a$ of 395 μmol mol$^{-1}$ or 545 μmol mol$^{-1}$
(depending on e$C_a$ treatment) with a saturating light of 1800 μmol m$^{-2}$ s$^{-1}$ and a flow rate of 500 μmol s$^{-1}$ with
temperature controlled to a constant based on the seasonal temperature. These data were used to estimate $J_{max}$
and $V_{cmax}$ at 25 ºC using the *fitaci* function in the *plantecophys* R package (Duursma, 2015), using the measured
temperature responses of $J_{max}$ and $V_{cmax}$ described in the previous paragraph to correct to 25 ºC.
Repeated gas exchange measurements were made on the same leaves in the morning and afternoon under
prevailing field conditions and saturating light (photon flux density = 1800 μmol m$^{-2}$ s$^{-1}$) on four occasions in
2013 ("diurnal"; Gimeno et al., 2016). To expand the diurnal dataset, we obtained the points from $A$-$C_i$ curves at
field $C_a$ and combined the two data sets. These data were used to estimate the $g_1$ parameter in the stomatal
conductance model (Eqn. 1) using the *fitBB* function in the *plantecophys* R package (Duursma, 2015). One $g_1$
value was fitted to the data from each treatment and date. The $g_1$ values were then regressed against $θ$ measured
in each treatment group to estimate the impact of soil moisture availability on leaf gas exchange, following Eqn.
2. The $g_1$ values were related to the nearest measurements of $θ$ (within two weeks without rain). Eqn. 2 was
fitted to this data set using the non-linear least squares method (Figure 3).
The dark respiration rate of foliage, $R_{dark}$, was measured at least three hours after sunset at a range of leaf
temperatures (14-60 °C) in February 2016 also with LiCor 6400. The temperature dependence of $R_{dark}$ was fitted
using non-linear least squared method to all of the measured data using Eqn. S6. Light responses of
photosynthesis were measured on two trees from each ring in October 2014 (Crous et al., unpublished). This
data set was used to constrain the light response parameters ($α_J$ and $θ_J$) in Eqn. S4. Details of fitting the light
response curves are provided in supplementary (Text S1).
**2.4 Model simulations and analysis**
MAESPA was used to simulate radiation interception and gas exchange of all six rings between 1 January 2013
and 31 December 2016 on a half-hourly basis. The model simulated half-hourly gross primary production (GPP)
of each tree, which was then summed for all trees in each ring to get the total annual GPP for each ring and year.
Four different sets of simulations were used to estimate carbon uptake under ambient and e$C_a$ and to identify the
key limiting factors on canopy GPP response to e$C_a$. Firstly, we carried out a simulation of leaf scale ("leaf
scenario") photosynthesis with measured meteorological data but fixed physiological data ($g_1$ = 3.3 kPa$^{0.5}$,
$V_{cmax.25}$ = 91 μmol m$^{-2}$ s$^{-1}$, and $J_{max.25}$ = 159 μmol m$^{-2}$ s$^{-1}$). This simulation aimed to quantify the $CO_2$ response of
Rubisco-limited and RuBP-limited photosynthesis at the leaf scale. This calculation was made using the





*photosyn* function in *plantecophys* R package (Duursma, 2015). This function implements the leaf gas exchange
routine used in MAESPA.
Secondly, MAESPA was run for all six rings with ambient $C_a$ and with $V_{cmax.25}$ from ambient measurements
("ambient scenario"). The results of this simulation were used to calculate the GPP of each ring under ambient
conditions. The ambient GPP values were also used to evaluate the inherent variability among the rings.
Thirdly, all six rings were simulated with $eC_a$ and $V_{cmax.25}$ based on measurements from ambient rings ("elevated
scenario"). The results of this simulation were compared to those from the ambient scenario to illustrate the
instantaneous response of canopy GPP to $eC_a$ in each ring and year. This simulation also quantifies the variation
of the GPP response to $eC_a$ across rings and years.
Lastly, we simulated the response of the three rings exposed to $eC_a$ (rings 1, 4, and 5) using the $V_{cmax.25}$ and $eC_a$
measured from these elevated rings ("field scenario"). Results from the field scenario were used for two
analyses: (i) to compare GPP from the field scenario to that of the three rings from the elevated scenario (i.e.,
$eC_a$ and ambient $V_{cmax.25}$), which allows us to quantify the impact of photosynthetic acclimation (i.e., due to a
reduction in $V_{cmax}$); (ii) to calculate the difference in GPP between the three ambient rings in ambient scenario
and elevated rings in the field scenario to estimate the response of GPP to $eC_a$ in the field.
*Table 1. Summary table of parameter definitions, units, and sources used in this study.*

| Parameters | Definitions | Units | Values | Eqn. |
|---|---|---|---|---|
| $\alpha_J$ | Quantum yield of electron transport rate | μmol μmol$^{-1}$ | 0.30 | S7 |
| $a$ | Fitted slope of LA and DBH | m$^2$ m$^{-1}$ | 492.6 | 4 |
| $a_{abs}$ | Absorptance of PAR | fraction | 0.825 | S4 |
| $b$ | Fitted index of LA and DBH | - | 1.8 | 4 |
| $c_D$ | Slope of $V_{cmax}$ to $D$ | kPa$^{-1}$ | 0.14 | 3 |
| $\Delta S$ | Entropy factor | J mol$^{-1}$ K$^{-1}$ | 639.60 ($V_{cmax}$); 638.06 ($J_{max}$) | S5 |
| $E_a$ | Activation energy | J mol$^{-1}$ | 66386 ($V_{cmax}$); 32292 ($J_{max}$) | S5 |
| $g_{1.max}$ | Maximum $g_1$ value | kPa$^{0.5}$ | 5.0 | 2 |
| $H_d$ | Deactivation energy | J mol$^{-1}$ | 200000 | S5 |
| $\theta_J$ | Convexity of electron transport rate to $Q_{APAR}$ | - | 0.48 | S8 |
| $\theta_{max}$ | Upper limit which $\theta$ has impact on $g_1$ | - | 0.240 | 2 |
| $\theta_{mim}$ | Lower limit which $\theta$ has impact on $g_1$ | - | 0.106 | 2 |
| $J_{max.25}$ | Value of $J_{max}$ at 25ºC | μmol m$^{-2}$ s$^{-1}$ | 159 | 3 |
| $k_T$ | Sensitivity of $R_{dark}$ to temperature | ºC$^{-1}$ | 0.078 | S6 |
| $q$ | The non-linearity of the $g_1$ dependence of $\theta$ | - | 0.425 | 2 |
| $R_{day.25}$ | Light respiration rate | μmol m$^{-2}$ s$^{-1}$ | 0.9 | S6 |
| $R_{dark.25}$ | Dark respiration rate | μmol m$^{-2}$ s$^{-1}$ | 1.3 | S6 |
| $R_{gas}$ | Gas constant | J mol$^{-1}$ K$^{-1}$ | 8.314 | S5 |
| $V_{cmax.25}$ | Value of $V_{cmax}$ at 25ºC | μmol m$^{-2}$ s$^{-1}$ | 91 (ambient); 83 (elevated) | 3 |




## 3. Results

Figure 4 summarises the results from measurements and the different simulations conducted in this study. It demonstrates that the impact of $eC_a$ diminishes as calculations are scaled from the instantaneous leaf-level response ($A_{inst}$) to the long-term canopy response ($GPP_{field}$) and the various feedback effects are accounted for. Each row of Figure 4 is explained in detail in the following paragraphs.

### 3.1 Instantaneous $C_a$ response of photosynthesis at leaf and canopy scale

The mean instantaneous $C_a$ response of leaf-level photosynthesis ($A_{inst}$) was +33% (Figure 4a). This response ratio was calculated from ~600 light- and temperature-controlled $A$-$C_i$ curves measured in the ambient rings. From the curves, we extracted the photosynthesis at 400 and 550 $C_a$ (μmol mol$^{-1}$) and calculated the instantaneous $C_a$ effect as their ratio. This approach allows an estimation of the direct $CO_2$ response independent of the impact of photosynthetic acclimation.

By contrast, the modelled direct GPP response to $eC_a$ was considerably less, just +11%, as shown in Figure 4d ("$GPP_{inst}$"). This canopy response rate was calculated by comparing the modelled GPP of all six rings under ambient and elevated $C_a$ ("ambient" vs. "elevated" scenario). As a result, this direct canopy GPP response also excludes the impact of photosynthetic acclimation.

Our results show that the major reason for the difference between the direct leaf and canopy photosynthesis responses to $eC_a$ is the relative contributions from Rubisco- and RuBP-regeneration-limited photosynthesis (cf. Figure 4 b and c). Figure 5 shows that the response of photosynthesis to $eC_a$ is considerably higher when Rubisco activity limits photosynthesis ($A_c$) than when RuBP-regeneration limits photosynthesis ($A_J$). When averaged over the range of leaf temperatures experienced during the four years of experiment, the $A_c$ response to $eC_a$ on average (+26%; Figure 4b) is larger than that of $A_J$ (+10%; Figure 4c). Leaf gas exchange measurements were taken in saturating light (1800 μmol m$^{-2}$ s$^{-1}$) and thus, are mostly Rubisco limited. The observed response rate of $A_{inst}$ is thus close to that of $A_c$.

At the canopy scale, a large fraction of the modelled canopy photosynthesis is limited by RuBP-regeneration. In Figure 6, we show the distribution of $A_c$ and $A_J$ during the four years of simulation as calculated by MAESPA. On average, 70% of the canopy photosynthesis is limited by RuBP-regeneration under ambient conditions ("ambient scenario"). The high fraction of $A_J$ is partly a consequence of the relatively low ratio of $J_{max.25}$ to $V_{cmax.25}$ (J:V ratio) which was estimated to be 1.7 (Table 1). In Figure 7, we estimated the PAR level at which Rubisco activity becomes limiting to leaf photosynthesis. The transition point from Rubisco- to RuBP-regeneration-limited photosynthesis was calculated from the leaf gas exchange sub-model by assuming a constant $C_a$ (390 μmol mol$^{-1}$), $D$ (1.5 kPa), $g_1$ (3.3 kPa$^{0.5}$), and $V_{cmax.25}$ (90 μmol m$^{-2}$ s$^{-1}$) but varying leaf temperature. As shown, under these conditions, when temperature = 25 °C and J:V ratio = 1.7, Rubisco activity limits photosynthesis only when incident PAR > 1800 μmol m$^{-2}$ s$^{-1}$. Using a higher J:V ratio such as the commonly-used value of 2 would decrease the saturating PAR value at which photosynthesis becomes Rubisco limited. We ran additional simulations assuming a J:V ratio of 2 and found that, with this ratio, MAESPA estimated 48% of photosynthesis to be RuBP-regeneration limited under ambient conditions and a direct GPP response of 15% (data not shown).





The shape of the light response curve also determines the transition point from RuBP- to Rubisco-limited
photosynthesis. We explored this effect by investigating the effect of varying the convexity, $\theta_J$. At EucFACE,
this parameter is estimated to be 0.48 based on data collected on site, indicating a shallow curvature and a high
light saturation points, in contrast to the commonly assumed 0.85, representing a steeper curvature and a lower
light saturation point. Using a value of 0.85 for $\theta_J$ resulted in a much lower PAR required for photosynthesis to
became Rubisco limited (dashed curves in Figure 7). With a $\theta_J$ of 0.85 and a J:V ratio of 1.7, MAESPA
estimated 40% of photosynthesis to be RuBP-regeneration limited under ambient conditions and a direct GPP
response of 16% (data not shown). With a $\theta_J$ of 0.85 and a J:V ratio of 2, MAESPA estimated just 34% of
photosynthesis to be RuBP-regeneration limited under ambient conditions and a direct GPP response of 18%
(Figure S2). The simulated $CO_2$ response of canopy carbon uptake thus depends heavily on the parameterisation
of light response and J:V ratio.

### 3.2 Acclimation of photosynthesis

The above calculations are made considering only the instantaneous response of photosynthesis to e$C_a$.
However, photosynthetic acclimation was observed at leaf scale (Ellsworth et al., in prep), and will also reduce
the response of GPP to e$C_a$ at the canopy scale. At the leaf-level, photosynthesis measured in the elevated rings
after five years of treatment ($A_{long}$) was 19% higher than that measured in ambient rings (Figure 4e; Ellsworth et
al. 2017). $A_{long}$ thus accounts for the photosynthetic acclimation in the elevated rings after four years of exposure
to e$C_a$. $A_{long}$ is considerably smaller than $A_{inst}$ (19% vs. 33%; Figure 4 a and e), indicating a large effect of
photosynthetic acclimation on the e$C_a$ response of light-saturated photosynthesis.
Accounting for the impact of photosynthetic acclimation in MAESPA, by using the $V_{cmax}$ from elevated rings
("field" vs. "ambient" scenarios) reduced the response of GPP to $C_a$ from 11% to 10% (GPP$_{long}$; Figure 4f). As
such, the photosynthetic acclimation had a relatively modest impact on the modelled annual GPP in the model.
The small impact of photosynthetic acclimation on canopy photosynthesis relative to the effect on leaf
photosynthesis can be explained by the fact that the leaf photosynthesis data are measured under saturating light
and thus are typically Rubisco-limited, so a reduction in $V_{cmax}$ had a large effect. In contrast, at the canopy scale,
much of the photosynthesis was limited by RuBP-regeneration and was largely unaffected by a reduction in
$V_{cmax}$.

### 3.3 Influence of LAI

The realised GPP response to e$C_a$ also depends on the canopy structure, specifically the LAI. In this experiment,
there was no significant change in LAI with e$C_a$ (-4% ± 5%; Figure 4g; see also Duursma et al. 2016). The
effect of e$C_a$ on LAI was calculated as the average effect between elevated and ambient annual mean LAI.
However, there was inherent variability in LAI across the rings (Figure 2a), which does not fundamentally
change the effect of e$C_a$ but requires a detailed analysis of the potential effects of natural variability on the
response to e$C_a$.



The small pre-treatment difference in LAI across rings gives rise to a range of estimates for the GPP response to
e$C_a$ in the field (6% ±8%; Figure 4h). This result is explored further in Figure 8, which combines the results
from "ambient", "elevated", and "field" scenarios. The average GPP across all six rings under ambient $C_a$ was
1574 g C m$^{-2}$ yr$^{-1}$ over the four-year simulation ("ambient scenario"; Figure 8). However, there was significant
variability in ambient GPP across rings, related in part to the inherent variability in LAI across rings. We
characterised the pre-existing differences in LAI by the initial LAI (LAI$_i$), measured on 26 October 2012. These
initial values are low, because they are measured immediately before the seasonal leaf flush, but characterise the
difference in LAI across rings over the full experimental period. Rings 1 and 4 (both experimental rings) have
the lowest LAI$_i$ (<0.3 m$^2$ m$^{-2}$) and thus the lowest average GPP under ambient conditions (1206 g C m$^{-2}$ yr$^{-1}$).
Ring 5 (the other experimental ring) has the second highest LAI$_i$ (~0.4 m$^2$ m$^{-2}$) and also the highest GPP under
ambient conditions (2359 g C m$^{-2}$ yr$^{-1}$). The variability among rings in ambient GPP (SD = 15%) is thus larger
than the modelled direct effect of $C_a$ on GPP, which is similar in all rings (+11%).
Owing to the variability among rings represented by LAI$_i$, the estimated mean GPP response to e$C_a$ across the
experimental rings has a sizeable confidence interval (±8%, Figure 4h). The actual e$C_a$ response was estimated
as an average effect between the ambient and elevated GPP values considering the impacts of photosynthetic
acclimation and inter-ring variability. The average GPP of experimental rings under field conditions (e$C_a$) was
estimated to be 1698 g C m$^{-2}$ yr$^{-1}$ while the average GPP of control rings under field conditions (ambient $C_a$)
was 1599 g C m$^{-2}$ yr$^{-1}$, an increase of 6% as shown in the Figure 4h. The variation of annual average GPP of the
control and experimental groups (blue and red squares in Figure 8) are thus represented by the CI in Figure 4h.

**4.  Discussion**
We have showed how a large response of leaf-level photosynthesis to e$C_a$ diminishes when integrated to the
canopy-scale, according to the synthesis of four years of leaf measurements at EucFACE with the stand-scale
model, MAESPA. We estimated that the canopy GPP of a mature *Eucalyptus* woodland under ambient $C_a$
conditions varied from 1084–2129 g C m$^{-2}$ yr$^{-1}$ by ring and year with a mean of 1574 g C m$^{-2}$ yr$^{-1}$. The model,
constrained by site measurements, predicted that once scaled to the canopy, the response of GPP to e$C_a$ only
increased by 6% (95% CI of ±8%) compared to the 19% (95% CI of ±5%) observed in leaf-scale measurements.
We were able to quantify the response of GPP to e$C_a$ and attribute the reduction in the response to various
factors including: (i) Rubisco versus RuBP-regeneration limitations to photosynthesis; (ii) photosynthetic
acclimation; (iii) inter-ring variability in LAI. Together these findings provide valuable insights into the relative
importance of each factor and help close a key knowledge gap in our understanding of how mature forests
respond to e$C_a$.
**4.1 Performance of MAESPA under ambient conditions**
The ambient GPP of EucFACE estimated by MAESPA was comparable to that measured with eddy covariance
in similar evergreen Eucalypt forests in Southeast Australia. In a nearby eddy covariance site (<1 km), ,
Renchon et al. (2018) estimated the ecosystem GPP from eddy convariance to be 1561 g C m$^{-2}$ yr$^{-1}$ during 2013
to 2016 which is within the range estimated for the ambient rings in this study, though this latter site and the
EucFACE are not the same in terms of canopy structure and LAI. Furthermore, our version of MAESPA was



evaluated against leaf photosynthesis and whole-tree sap flow measurements in EucFACE ($R^2$ of 0.77 and 0.8,
respectively; Yang et al., in review). These comparisons indicate MAESPA is a useful tool to explore the
canopy carbon uptake and the predicted GPP could provide a baseline to future studies.
**4.2 RuBP-regeneration limited photosynthesis**
Our results show that the canopy GPP at EucFACE was predominantly limited by RuBP regeneration. The
reason for the frequent RuBP-regeneration limitation is that the measured J:V ratio was relatively small in
EucFACE (1.7), and stomata tend to close at midday when light levels are higher and Rubisco-limitation is
expected (Gimeno et al., 2016). A lower J:V ratio increases the PAR threshold required for the photosynthesis
model to switch between the RuBP-regeneration limitation and the Rubisco limitation (from <1000 to <1800
$\mu$mol m$^{-2}$ s$^{-1}$; Figure 7). Previous studies have highlighted the need to consider J:V ratio for a correct prediction
of $CO_2$ response (Long et al, 2004; Zaehle et al., 2014; Rogers et al., 2017). However, as shown by Zaehle et al.
(2014), Medlyn et al. (2015), and Rogers et al. (2017), current models differ in their predictions of the transition
from RuBP-regeneration- to Rubisco-limited photosynthesis, suggesting the uncertainty of predicted $CO_2$
response of GPP could be reduced by using a realistic J:V ratio.
Previous modelling studies applying MAESPA to e$C_a$ experiments both assumed higher J:V ratio (2) and
estimated higher GPP response to e$C_a$ presumably due to less frequent RuBP-regeneration limitation (Wang et
al., 1998; Luo et al., 2001). A J:V ratio of 2 was suggested by Wullschleger (1993) and has been used in many
modelling studies (e.g., the seven terrestrial biosphere models assessed by Rogers et al. (2017) all assumed a J:V
ratio of 1.9-2). Global terrestrial biosphere models such as JULES and others frequently estimate $J_{max}$ on the
basis of this ratio (e.g., Clark et al. 2011). However, the relatively low J:V ratio observed at EucFACE is not
unique. In the Duke Forest FACE site in the US, Ellsworth et al. (2012) reported a J:V ratio of ~1.7 which is the
same as that estimated for EucFACE. Kattge and Knorr (2007) analysed $V_{cmax}$ and $J_{max}$ values from 36 species
across the world and found a low J:V ratio (<1.8) in herbaceous, coniferous, and broadleaved species. Most
recently, Kumarathunge et al. (2018) studied the variation in J:V ratio in datasets obtained from around the
globe and found a consistent relationship with growing season temperature. The ratio varied from 2.5 in tundra
environments to < 1.5 in tropical environments. The value of 1. 7 observed at EucFACE falls within this
prediction for the prevailing growth temperature at this site. The inclusion of his relationship between this
relationship of J:V ratio and temperature will thus be important for capturing the GPP response to e$C_a$ globally.
We also found that the curvature of the light response of photosynthesis affected the predicted GPP response to
e$C_a$ (Figure 7). The parameter value we fitted to data measured *in situ* ($\theta_J$ = 0.48) is lower than the value
commonly assumed in the models (typically around 0.85, e.g. Medlyn et al., 2002; Harverd et al., 2018).
Nonetheless, our relatively low $\theta_J$ value (<0.7) is not unique, as it is also supported by a number of studies on
different species around the world (Ögren, 1993; Valladares et al., 1997; Lewis et al., 2000; Hjelm and Ögren,
2004). The inclusion of higher $\theta_J$ value would predict a much higher direct GPP response to e$C_a$ (e.g., 16%
versus 11% in this study), because higher $\theta_J$ results in a large proportion of GPP being Rubisco-limited. This
finding calls for careful examination of the light-response of photosynthesis, which has a large effect on the
predicted e$C_a$ response



### 4.2 Photosynthetic acclimation

Some degree of photosynthetic acclimation (i.e., a long-term reduction of $V_{cmax}$ under $eC_a$) has been widely reported in FACE studies and has been attributed to a reduction of leaf nitrogen concentration (Saxe et al., 1998; Ainsworth and Long, 2005). The response of GPP to $eC_a$ would be linearly related to $V_{cmax}$ if photosynthesis were mostly limited by Rubisco activity. Photosynthetic acclimation was responsible for the reduced response of leaf-scale light-saturated photosynthesis from 33% ($A_{inst}$) to 19% ($A_{long}$). However, this reduction in $V_{cmax}$ translated into only a ~2% reduction in GPP modelled by MAESPA. Wang et al. (1998) also showed that photosynthetic acclimation (-21% in $V_{cmax}$) reduced modelled canopy GPP by only 6% due to RuBP-regeneration being the primary limitation of canopy photosynthesis. These findings thus suggest that photosynthetic acclimation may only have a small effect in the GPP response to $eC_a$ when canopy photosynthesis is mostly RuBP-regeneration limited. This response is thus consistent with the hypothesis that the reduction in $V_{cmax}$ represents a re-allocation of nitrogen to optimise nitrogen use efficiency under $eC_a$ (Chen et al., 1993; Medlyn et al., 1996).

### 4.3 Constraining the carbon balance response to $eC_a$

At EucFACE, after four years of $eC_a$ treatment, there was no evidence of increased above-ground tree growth (Ellsworth et al., 2017). Nor have the trees at EucFACE shown any significant change in LAI (Duursma et al., 2016). The relatively small response of GPP and the effect of ring-to-ring variation provides important context for these statistically non-significant responses of tree growth at the stand scale at EucFACE. Firstly, the effect size calculated for GPP of +11% (+ 169 g C $m^{-2}$ $yr^{-1}$) constrains the likely effect size for plant growth and other components of the ecosystem carbon balance and is a more useful baseline for comparison than the response of light-saturated leaf photosynthesis (+19% = 299 g C).

Secondly, the inherent ring-to-ring variation in this natural forest stand is even higher than the GPP response, which highlights the importance of considering both the effect size and uncertainty than to focus on statistical significance. It is important to note that the EucFACE site could be considered relatively homogeneous for a mature woodland. The site is flat, trees appear similar-aged, and almost all the overstory belongs to a single species. In addition, plots were carefully sited to minimise variation in basal area. However, there are small-scale variations in soil type, depth, and nutrient availability that cause variation in LAI. This scale of variation is likely to present in other natural forests, and indeed, other studies on mature trees also note that background variability can contribute to the lack of statistically significant findings (Fatichi and Leuzinger, 2013; Sigurdsson et al. 2013). We highlight the need to focus on effect size and its uncertainty, rather than the dichotomous significant/non-significant approach when evaluating experimental results from native forests.

### 4.4 Implications for terrestrial biosphere models

Seven Terrestrial Biosphere Models (TBMs) were used to predict GPP and LAI responses to $eC_a$ in advance of the EucFACE experiment (Medlyn et al. 2016). The predicted $eC_a$ responses of GPP ranged from +2 to +24% across the seven models, while the predicted responses of LAI ranged from +1 to +20%. With our results, it is possible to falsify some of these model simulations. The model with the lowest GPP response (CLM4-P) assumed very strong down-regulation of photosynthesis owing to phosphorus limitation. However, this down-regulation was not observed here. The models with the highest GPP responses (GDAY, O-CN, SDGVM) had a





J:V ratio of 2 which is higher than that observed at EucFACE, and also had a positive feedback to GPP via
increased LAI (+5-15%), which did not occur (Duursma et al., 2016). The model rendering most similar
prediction for the GPP response to e$C_a$ to the output of MAESPA incorporating empirical observations was the
CABLE model. This latter model predicted an e$C_a$ response of GPP of ~12% with a large proportion of RuBP-
regeneration limited photosynthesis, both of which are similar to the findings in this study. Future TBMs may
benefit from incorporating a more realistic representation of the relative contribution of RuBP-regeneration- to
Rubisco- limited photosynthesis to GPP. For instance, adding the temperature dependency of J:V ratio could
help capture the variation of J:V ratio globally (e.g., Kumarathunge et al., 2018).
Our study provides a number of process-based insights that can be used to improve model performance both
qualitatively and quantitatively. Our modelling exercise is also a major contribution to the understanding of the
EucFACE experiment by quantifying the amount of extra carbon input into the system by canopy-level
photosynthesis and thus providing a reference for assessing the impacts of e$C_a$ on growth and soil respiration.
Finally, our study highlights that the e$C_a$ effect on canopy-scale GPP may be considerably lower than the effect
on photosynthesis of the light-saturated leaves, due to contrasting relative limitations to photosynthesis
operating and different scales. In future work, our GPP estimates will be used as an input to calculate the overall
effect of e$C_a$ on the carbon balance at the whole EucFACE site.
**Acknowledgements**
JY was supported by a PhD scholarship from Hawkesbury Institute for the Environment, Western Sydney
University. MGDK was supported by NSW Research Attraction and Acceleration Program (RAAP).
EucFACE was built as an initiative of the Australian Government as part of the Nation-building Economic
Stimulus Package and is supported by the Australian Commonwealth in collaboration with Western Sydney
University. It is also part of a TERN Super-site facility.
We thank Vinod Kumar, Craig McNamara and Craig Barton, for their excellent technical support. We also
thank Elise Dando for help in measuring crown radius, Steven Wohl for crane driving, Julia Cooke and Burhan
Amiji for installing the neutron probe access tubes.
**Author contribution statement**
JY, BM, MDK, and RD conceived and designed the analysis. KC, DE, and TG designed sampling of leaf
physiological data, while DE and RD designed sampling of canopy structure data. KC, DE, TG, AWK, RD and
JY collected data. RD and DK provided analysis tools. JY and BM performed the analysis. JY, BM, MDK, and
MJ wrote the paper. All authors edited and approved the manuscript.

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





**Figures and Captions**


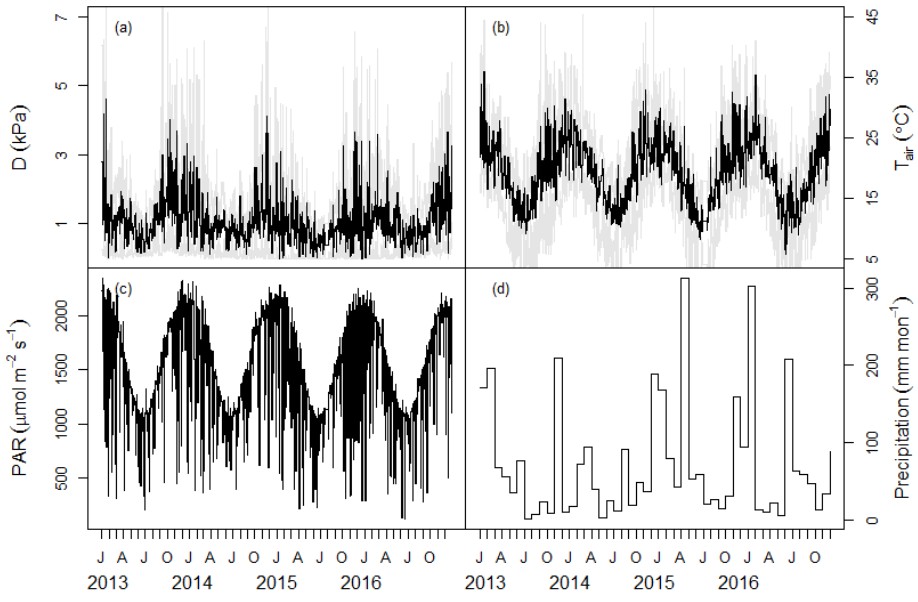


*Figure 1. Meteorological data measured at the site during the period 2013-2016. Panels show (a) daily mean*
*vapour pressure deficit (D) with shaded area marking the maximum and minimum of the day, (b) daily mean air*
*temperature ($T_{air}$) with shaded area marking the maximum and minimum of the day, (c) daily maximum*
*photosynthetically active radiation (PAR), and (d) monthly total precipitation. Note that precipitation has no*
*direct impact in the model but modifies stomatal conductance via the change in soil moisture.*

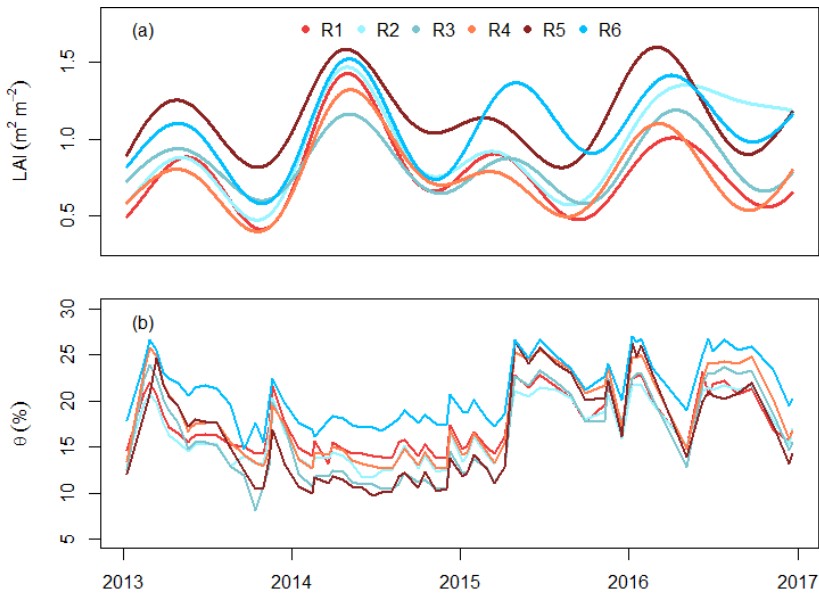

*Figure 2. (a) Leaf area index (LAI) and (b) volumetric water content (θ) used to drive the model. LAI was measured in each ring using the measured absorbed PAR and smoothed using generalized additive model following Duursma et al. (2016). θ was measured using neutron probes at top 150 cm biweekly and gap-filled using a linear interpolation between two nearest available data (Gimeno et al. 2018).*

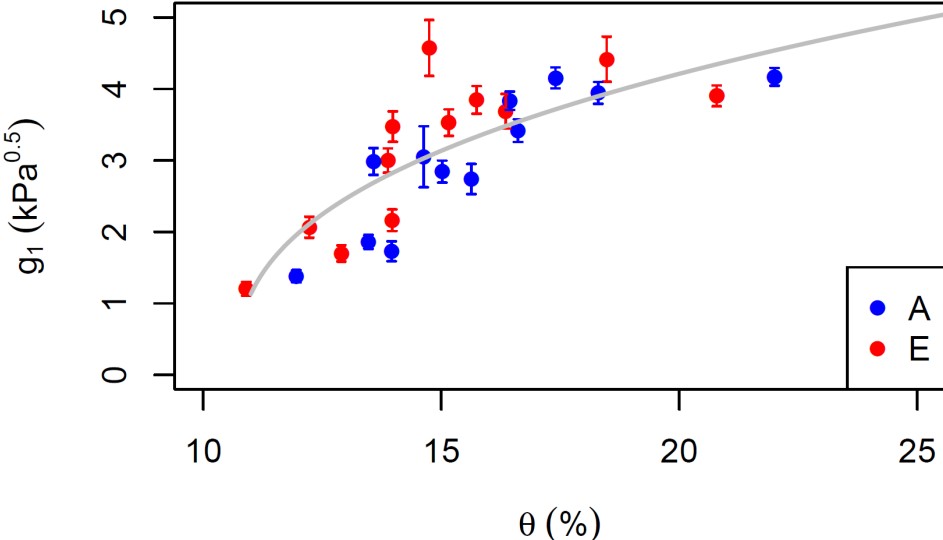

*Figure 3. The impact of soil moisture content (θ) at top 150 cm on stomatal regulation. Red dots are fitted to data from elevated rings while blue are ambient rings. The bars mark the standard errors of the fitted values. The grey line shows the fit of Eqn. 2 to the data.*



716

717

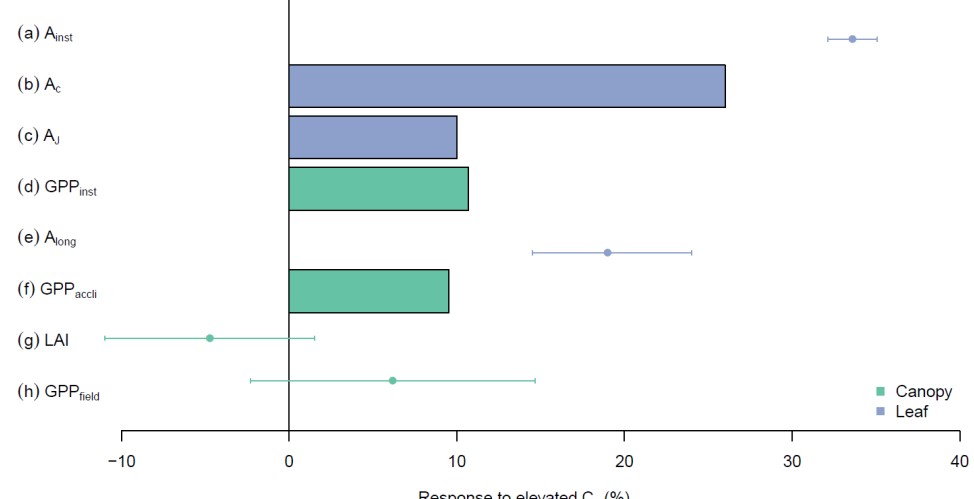

718

*Figure 4. The response of photosynthesis to $eC_a$ on different scales and limited by different factors. In summary, from top to bottom, the figure demonstrates how a large increase in leaf photosynthesis can diminish into a non-statistically significant change in canopy GPP under $eC_a$. Entries from top to bottom are as follows. (a) $A_{inst}$, the instantaneous response of leaf photosynthesis to $eC_a$ obtained from $A$-$C_i$ measurements in ambient rings (error bars indicate 95% CI). (b) $A_c$, the modelled response of Rubisco-limited leaf photosynthesis, assuming no down-regulation, averaged over the range of diurnal air temperatures experienced during the experimental period. (c) $A_J$, the modelled response of RuBP-regeneration limited leaf photosynthesis. (d) $GPP_{inst}$, the direct effect of $eC_a$ on canopy GPP, modelled with MAESPA, assuming no downregulation of photosynthesis and averaged across all six rings. (e) $A_{long}$, the long-term response of leaf photosynthesis to $eC_a$ obtained from leaf photosynthesis measured at treatment $CO_2$ concentrations (see Ellsworth et al. 2017). This value is different from $A_{inst}$ because it incorporates photosynthetic acclimation. (f) $GPP_{long}$, the effect of $eC_a$ on canopy GPP once the measured down-regulation of $V_{cmax}$ is taken into account. (g) LAI, the measured difference in average LAI between $eC_a$ and ambient $C_a$ rings over the experiment period (data from Duursma et al. 2016). (h) $GPP_{field}$, the GPP response modelled with MAESPA comparing the three elevated rings with the three ambient rings. See text for further explanation.*






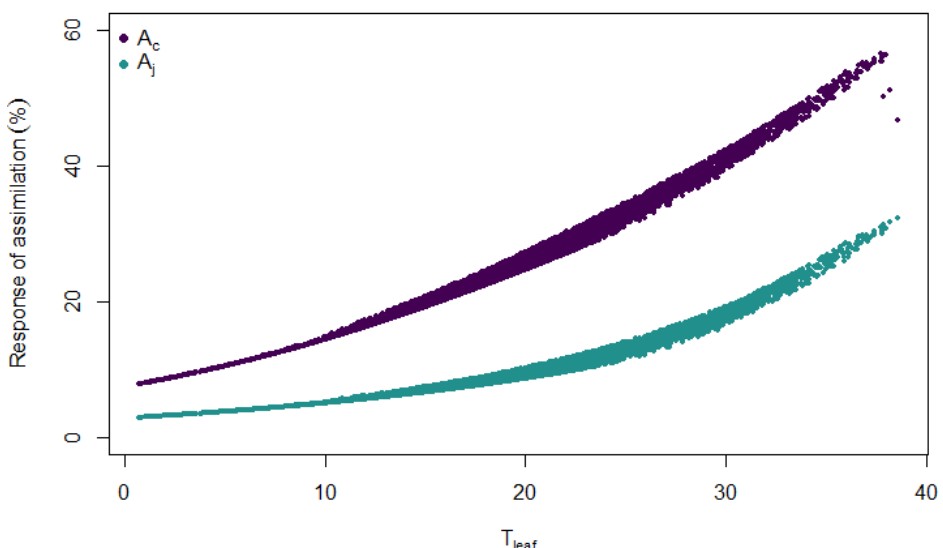


*Figure 5. The modelled $C_a$ response of Rubisco-limited leaf photosynthesis ($A_c$) and RuBP-regeneration-limited*
*leaf photosynthesis ($A_J$) against leaf temperature ($T_{leaf}$). The responses are calculated for temperatures during*
*the period 2013-2016. Parameters are as given in Table 1, except that $V_{cmax.25}$ and $g_1$ were assumed to be*
*constant for clarity ($g_1 = 3.3$ kPa$^{0.5}$ and $V_{cmax.25} = 90$ $\mu$mol m$^{-2}$ s$^{-1}$).*

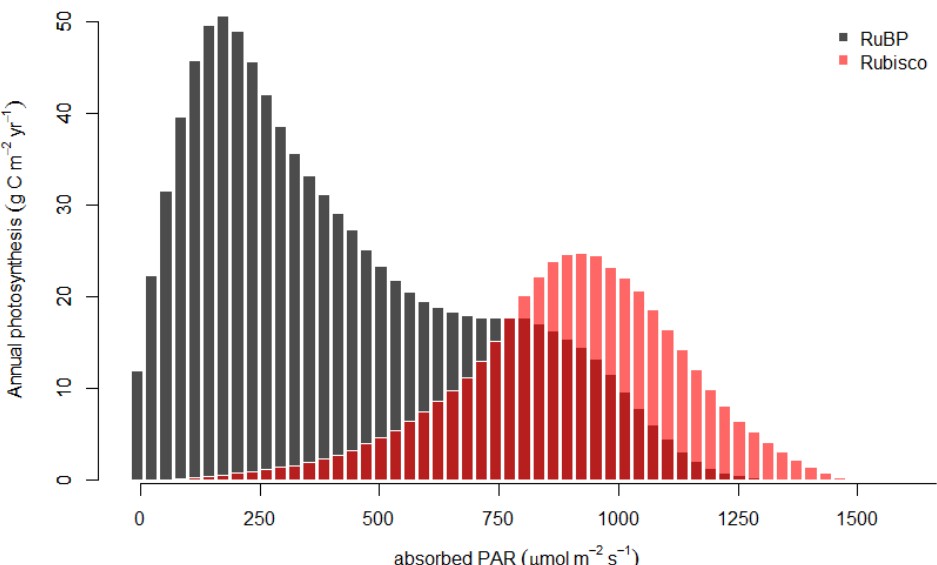




*Figure 6. Distribution of average annual photosynthesis limited by Rubisco activity and RuBP-regeneration in*
*bins of absorbed PAR (25 μmol m$^{-2}$ s$^{-1}$)., as calculated by MAESPA across all rings during 2013-2016. The*
*histogram was constructed by calculating the photosynthesis (either limited by Rubisco or RuBP) falling into*
*each bin for every half-hour in the "ambient scenario". These values were then summed to each year and ring*
*and averaged over six rings and four years.*

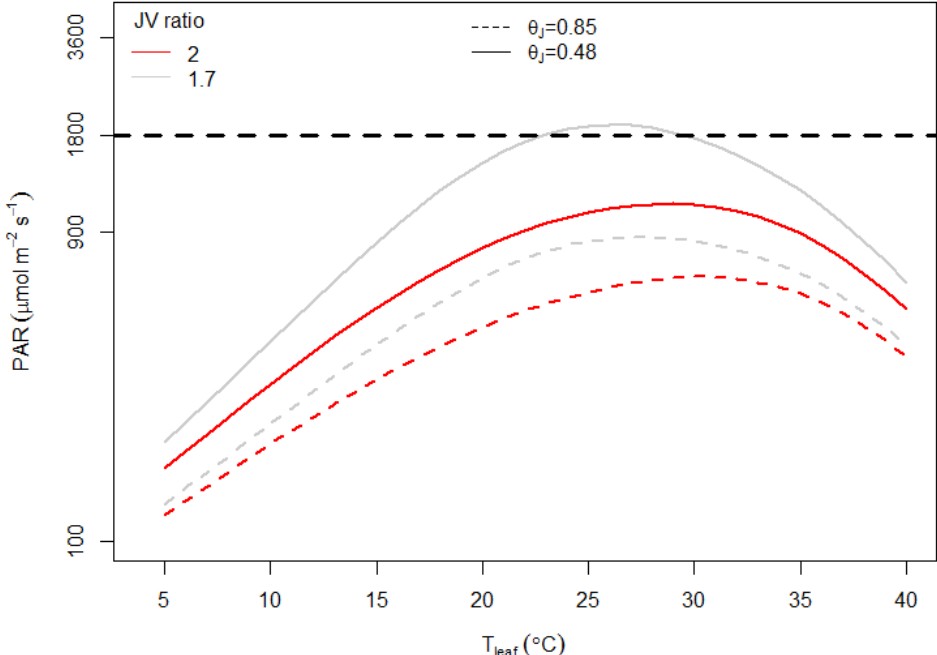


*Figure 7. Estimated PAR value at which limitation to photosynthesis shifts from RuBP generation to Rubisco at*
*different leaf temperatures and J:V ratios. Rubisco limitation occurs at PAR values above the curves; RuBP*
*regeneration limitation occurs below the curves. The curves were calculated using the Photosyn function in the*
*plantecophys R package (Duursma, 2015). The parameters other than PAR and $T_{leaf}$ were assumed to be*
*constant: $C_a$ = 390 μmol mol$^{-1}$; D =1.5 kPa; $g_1$ = 3.3 kPa$^{0.5}$; $V_{cmax.25}$ = 90 μmol m$^{-2}$ s$^{-1}$. The temperature and*
*light dependences of photosynthesis were assumed to be the same as in MAESPA. The grey line was predicted*
*by assuming $J_{max.25}$ = 153 μmol m$^{-2}$ s$^{-1}$ (i.e., J:V ratio= 1.7). This J:V ratio was observed consistently in*
*EucFACE across campaigns and rings. The red line was predicted by assuming $J_{max.25}$ = 180 μmol m$^{-2}$ s$^{-1}$ (i.e.,*
*J:V ratio= 2). This J:V ratio was commonly reported and used in other studies. The horizontal dashed line*
*shows the PAR = 1800 μmol m$^{-2}$ s$^{-1}$ at which leaf-level measurements of EucFACE were made. Note the log*
*scale of the y axis. The dashed curved are based on quantum yield of electron transport ($\alpha_J$; mol mol$^{-1}$) and*
*(Convexity of light response of RuBP; $\theta_J$ ; unitless)values from CABLE model (Haverd et al., 2018).*
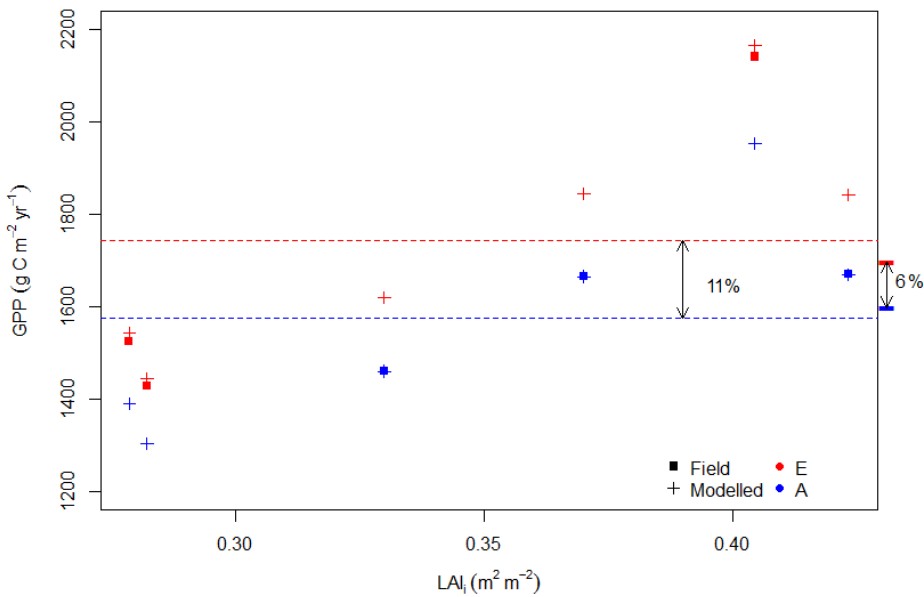

*Figure 8. The four-year average GPP of all six rings under ambient and eC$_a$ plotted against initial leaf area*
*index (LAI$_i$). LAI$_i$ is the LAI measurement taken on the 26 October 2012 and is a proxy of the inherent variation*
*among the rings. For all six rings, estimated GPP is shown for ambient C$_a$ (blue) and eC$_a$ (red). Crosses*
*indicate GPP from simulations by varying C$_a$ and squares indicate GPP as under field conditions. The flat bars*
*on the right hand-side of the plot indicate the average ambient C$_a$ GPP for ambient rings only (the average of*
*blue squares) and average eC$_a$ GPP for elevated rings only (the average of red squares). Dashed lines indicate*
*average ambient C$_a$ (the average of blue crosses) and eC$_a$ GPP across all six rings (the average of red crosses).*
*The flat bars thus mark the modelled response without inter-ring variability while the dashed lines mark the*
*modelled realized response, including inter-ring variability.*