# Peer review of "Low sensitivity of gross primary production to elevated CO2 in a mature Eucalypt woodland 2 Authors: Jinyan Yang1, Belinda E. Medlyn1, Martin G. De Kauwe2,3, Remko A. Duursma1, Mingkai Jiang1, Dushan Kumarathunge1, Kristine Y. Crous"

_Biogeosciences, 2019_

## Referee Comment (RC1) · Simone Fatichi (Referee) · 24 Aug 2019

Overall Review

The article presents a detailed exercise of upscaling photosynthesis from the leaf scale to the stand level for the climatic conditions and vegetation distribution corresponding to the EucFACE experiment (forest stand dominated by Eucalyptos tereticornis) and thus it includes ambient and elevated CO2 (eCO2) scenarios. Upscaling leaf-level response to tree and forest stand scale is a long-standing problem in biogeoscience and while it has been tackled in various ways in the literature, the study presented here is innovative for the thoroughness and level of detail included in the analysis. Furthermore, the analysis is carried out for ambient and eCO2 conditions using a terrestrial

biosphere model (MAESPA) that represents explicitly each tree and solve the canopy using multiple layers and accounts in each layer for multiple points representing radial variability in incoming light. The study also accounts for the acclimation response of photosynthesis and it is strongly constrained by observations, which is rarely the case in other similar studies. The study convincingly shows that a strong increase in leaf-level light-saturated photosynthesis (+33%) under eCO2 reflects in a minor increase in stand level GPP (10%) because of the prevalence of electron transport limitations in photosynthesis and to a minor extent downregulation of photosynthetic capacity due to the leaf acclimation to eCO2. Results also show a large uncertainty in computing GPP at the stand level when a small area (corresponding to a CO2 enrichment ring) is considered. While upscaling photosynthesis at the forest stand scale is not a new task, the way this problem is solved here, represents a scientific advancement because it is presented in the context of a FACE experiment and provide a number of interesting discussion points on mechanistic model parameterization and uncertainties (e.g., the role of the curvature for electron transport, the Jc,max/Vc,max ratio, photosynthesis acclimation, forest stand heterogeneity). It is clearly shown that translating leaf-level responses of CO2 effects to the ecosystem scale is very misleading and most important the study provides mechanistic explanations for the differences. The search for the reasons and the clear explanations provided concerning subcomponents of the photosynthesis model (e.g., Rubisco vs. electron transport limited, or acclimation of photosynthetic capacity) represents an innovative approach, which I did not see before in the literature. For these reasons, beyond the importance of estimating GPP in ambient and eCO2 conditions that will serve future studies in the context of the Euc-FACE experiment, the article represents an important piece of work for the mechanistic understanding of ecosystem responses to elevated CO2.

The article is overall very well written and presented. In summary, I think the manuscript is making an important contribution to the field and I sincerely congratulate the authors for this nice piece of work. In the following, I just have a number of minor comments that can be helpful to improve further the presentation of this work.

Sincerely,

Simone Fatichi

Minor comments

P.2 Line 34-37. The difference between canopy scale "direct response" of +11% and the mean actual response of 6%, while very clear in the manuscript, it is not so clear at the abstract level. Maybe introducing the concept of "uncertainty" associated with the variability across rings or something associated to the "actual field response" according to the experimental configuration may help.

P.3 Line 51. The "hence" here is out of place, because the causality is not straight-forward. An increase in carbon uptake does not necessarily lead to an increase in the amount of carbon stored in the ecosystem. The authors are well aware of this. Something like "which in turn could potentially increase..." will be more correct.

P.3. Line 57. A short overview of main disagreements between various studies is provided in Fatichi et al. 2019.

P.3. Line 57-68. I think this paragraph would benefit from referring to the estimates of global terrestrial C sink. While the attribution of the land C-sink is still debated, an average C-sink of 20–30 g C year-1 m-2 over vegetated land in the last decades is not a detail in the overall story about eCO2.

P. 4. Line 80. While, practically, I would agree in defining the response of GPP to eCO2 an upper bound. Theoretically, this is not a limit, if for some reason, plants in eCO2 conditions will be able to do maintenance with half of the respiration costs, then the NPP response could be larger than the GPP response. I think a "reference value" is more correct than an "upper-bound".

P.4. Line 115 and P.6 Line 161 and 166. Yang et al. 2019 is missing from the reference list, overall, I would avoid referring to papers, which are not published.

P.4 Line 116 and P.6 Line 170-171. I would not mix the "meteorological forcing" with the "model parameterization". The two aspects are different from a modeling perspective, one represents the inputs to the model, the other (e.g., physiological and structural attributes) represents model parameters, or prognostic variables if these are time-dynamics and computed in the model. One can use the same model parameterization with different meteorological inputs and the other way around.

P.6 L.173. Figure 2b. I am strongly encouraging to avoid using a linear interpolation for soil moisture values at least in its graphical representation. Soil moisture temporal dynamics have a fast and strong response to rainfall events. Linearly interpolating biweekly value is creating a misleading perception of the real temporal dynamics of soil moisture. I would prefer to have just the points when the soil moisture values have been collected rather than the current representation where raising and descending soil moisture dynamics are often unrealistic.

P.7. Line 220. Just a suggestion. Maybe the Fig. S1 could be included in the main manuscript.

P.9 Line Table 1. I know that in literature it is quite typical to report $\mu$mol only. However, this is not very precise, especially when we are dealing with photosynthesis. I would suggest to explicitly say $\mu$mol of what, e.g., $\mu$mol-CO2, $\mu$mol-H20, $\mu$mol-electrons or better $\mu$mol-Eq. as in the original Farquhar et al. 1980.

P.10. Line 286-290. Did you check if with MAESPA you get the same +33% of leaf-level photosynthesis if you simulated the same environmental conditions of the 600 A-Ci curves? Very likely, yes, because these are used to estimate the photosynthesis parameters, but just as a double check.

P.10 Figure 6 and 7. In Fig.7 is reported incident PAR and in Fig. 6 absorbed PAR, even though one refer to the stand scale and the other to the leaf-level, I think it would have been better to use either absorbed or incident PAR in both of them for comparison.

P. 11. Line 318. From the Supp. Material, the curvature for electron transport $\theta$j is also used as curvature and for overall photosynthesis (Eq. S8). These two values are typically different in models (e.g., Bonan et al 2011). This needs to be specified in the manuscript as well. The reference $\theta$j =0.85 is typically assumed for the curvature and for the overall photosynthesis, rather than for the curvature of electron transport, which is typically lower in some models (0.7, Bonan et al 2011, Fatichi et al 2016). This needs to be discussed.

P. 12. Line 377. I am honestly impressed by the inter-ring differences in GPP. I think these are mostly related to the relative small size of the rings. Or better, the size is quite large in comparison to experimental capabilities but relative small to average forest stand heterogeneities.

P.12 Line 388. Renchon et al 2018 is not in the reference list.

Eq (S3) The denominator should be Ci+2$\Gamma$ rather than Ci+ $\Gamma$ (e.g., Wang and Leuning, 1998, Dai et. 2004, Bonan et al 2011);

References

Bonan, G. B., P. J. Lawrence, K. W. Oleson, S. Levis, M. Jung, M. Reichstein, D. M. Lawrence, and S. C. Swenson (2011), Improving canopy processes in the Community Land Model version 4 (CLM4) using global ux _elds empirically inferred from FLUXNET data, Journal of Geophysical Research, 116 (G02014), doi:10.1029/2010JG001593

Dai, Y., R. E. Dickinson, and Y.-P. Wang (2004), A two-big-leaf model for canopy temperature, photosynthesis, and stomatal conductance, Journal of Climate, 17, 2281-2299.

Farquhar, G. D., S. V. Caemmerer, and J. A. Berry (1980), A biochemical model of photosynthetic CO2 assimilation in leaves of C3 species, Planta, 149, 78-90.

Fatichi S., C. Pappas, S. Leuzinger, J. Zscheischler (2019). Modelling carbon sources and sinks in terrestrial ecosystems. New Phytologist, 221:652–668,

doi:10.1111/nph.15451

Fatichi S., S. Leuzinger, A. Paschalis, J. A. Langley, A. Donnellan Barraclough, and M. Hovenden (2016). Partitioning direct and indirect effects reveals the response of water limited ecosystems to elevated CO2. Proceedings of the National Academy of Sciences USA, 113(45) 12757-12762, doi: 10.1073/pnas.1605036113

Wang, Y.-P., and R. Leuning (1998), A two-leaf model for canopy conductance, photosynthesis and portioning of available energy I: Model description and comparison with a multi-layered model, Agricultural and Forest Meteorology, 91, 89-111.

---

## Referee Comment (RC2) · Anonymous Referee #2 · 11 Nov 2019

This manuscript syntheses a large amount of data from the EucFACE project to examine the effects of Rubisco- versus RuBP limitation on photosynthesis under elevated CO2. The authors present leaf-level measurements, leaf-level modeling, and canopy scale modeling of ambient versus elevated CO2 conditions to illustrate that current projections of GPP under elevated CO2 are overestimated in mature forests due to biases towards light-saturated leaves. This work is scientifically relevant and pedantic. I want to commend the authors on their efforts and have minor suggestions to improve the presentation and make the work clearer to a wider audience.

The introduction is extremely well written and provides appropriate context for the work being conducted within the manuscript. The methodology is thoughtfully presented, and justification was given for parameter choices in the model. The amount of data

used to represent the system is commendable and I appreciate the attention to detail. I find the presentation of soil moisture to be the weakest element of the methodology and would recommend a little more attention paid to it as it is one of the few varying parameters between the replicates. The presentation of the results would be strengthened by more clearly delineating measurements vs. leaf scale modeling vs. canopy scale modeling. I would personally be very interested in seeing some of the rawer data forms (e.g., timeseries of canopy model) in addition to the synthesized percent changes. While this may be a question of style, I found the figure captions to contain relevant information that was missing from the text. I would include more of that information in the text for clarity. Figures are adequate, but the figure legends are not descriptive (esp. Fig 2 and 4-7) and the long captions make it difficult to distinguish between the different replicates, responses, etc.

Overall well done and I'm excited to see this work published!

INTRODUCTION L94 -95 Can you please give an example of the ranges of Jmax:Vmax ratio found in these cited works to show how much it deviates from the normally adopted ratio of 2?

METHODS L132: Is the repo unchanged or should the reader be directed to a certain commit version?

L162: You do not introduce the variable D until line 172 and do not provide units.

L164: Can you please clarify the choice between Jmax and Vcmax here in the Vmax, t parameter?

L175-177: I would introduce the equipment and measurement heights before the frequency, but this is a minor point.

L186: In Fig 2 you present that you use 150cm neutron probe measurements that were conducted biweekly and linearly interpolated, but here you say these measurements were not gap-filled? While I do not think this would majorly affect your results, averaging

the soil moisture over the entire 150 cm profile seems problematic as you are giving equal weight to regions that are likely to contain significantly less root biomass. Would it not be more fitting to use a weighted average based on the below-ground biomass distribution to represent the soil moisture that the tree actually "feels"? Would this have changed your g1?

L202: Minor point of convention – normally see DBH represented in [cm]. I assume you used DBH in [cm] in your allometry in Eq 4?

L214-215: Fit statistics of this allometry?

L225: No, no to citing an "in prep" when you seem to be presenting this data in this work.

L278 Misspelling of \theta_min

L246: Please expand up on the statement "within two weeks without rain" – was there some selection of points that happened based on this? I'm bit confused with the g1 and soil moisture match up.

L249: This is quite the range of leaf temperatures indeed!

L255: Missing commas.

RESULTS The results presentation is somewhat difficult to follow given the large number of simulations and measurements spanning scales. Figure 4's mix of bar and point measurements is difficult to follow. Separating data measurements from modeled responses could help the reader follow better what is derived from models and what was an actual measured response. I appreciate the color coding between leaf and canopy measurements. Also, making a clearer distinction between the leaf-level models using the R package and the canopy scaled measures made using MAESPA would be helpful.

DISCUSSION

L397-398: I would love to see the timeseries that illustrates this stomatal closure at the canopy scale.

L405-418: Ah ha! This was the presentation of J:V ratio ranges that I was waiting for. I would still suggest adding in a range to the introduction so that the reader is primed to consider how variability in J:V could impact these modeled responses to eCa.

L420: Yes, but would how would the way you averaged your soil moisture values affect this value?

L449-451 I agree with this statement, but can you be more clear about what "uncertainty" you are referring to? Are you talking about uncertainty in our forcing variables for models; structural uncertainty in the models; both?

L462-463 Falsify those model simulations? Should we just throw the models in the trash or can we focus on an improvement in the model structure in order to capture these transitions between Rubisco limitation and RuBP regeneration? Or could it be also that there are other structural differences between those models and the explicit canopy structure of MAESPA?

FIGURES

L708-711 Fig 2: You give no clear indication about what the different line colors mean. I assume these are replicates, but I am not clear about if these are elevated or control plots. You briefly mention ring numbers in text, but the figure would be improved if you make this distinction more visually apparent.

L714: "error bars represent standard error of fitted values" I'm a bit confused by this statement.

L717: This figure is especially hard to follow and the mix of bar and points is difficult. I would suggest adding further groups to help identify measurements vs. leaf scale modeling vs. canopy scale modeling.

---

## Author Comment (AC1) · 30 Nov 2019

We would like to thank both reviewers for the positive and constructive comments. Their suggestions have helped improve our study. Our responses are listed below in blue and italic.

Response to reviewer 1

Overall Review

The article presents a detailed exercise of upscaling photosynthesis from the leaf scale to the stand level for the climatic conditions and vegetation distribution corresponding to the EucFACE experiment (forest stand dominated by Eucalyptos tereticornis) and thus it includes ambient and elevated CO2 (eCO2) scenarios. Upscaling leaf-level response to tree and forest stand scale is a long-standing problem in biogeoscience and while it has been tackled in various ways in the literature, the study presented here is innovative for the thoroughness and level of detail included in the analysis. Furthermore, the analysis is carried out for ambient and eCO2 conditions using a terrestrial biosphere model (MAESPA) that represents explicitly each tree and solve the canopy using multiple layers and accounts in each layer for multiple points representing radial variability in incoming light. The study also accounts for the acclimation response of photosynthesis and it is strongly constrained by observations, which is rarely the case in other similar studies. The study convincingly shows that a strong increase in leaflevel light-saturated photosynthesis (+33%) under eCO2 reflects in a minor increase in stand level GPP (10%) because of the prevalence of electron transport limitations in photosynthesis and to a minor extent downregulation of photosynthetic capacity due to the leaf acclimation to eCO2. Results also show a large uncertainty in computing GPP at the stand level when a small area (corresponding to a CO2 enrichment ring) is considered. While upscaling photosynthesis at the forest stand scale is not a new task, the way this problem is solved here, represents a scientific advancement because it is presented in the context of a FACE experiment and provide a number of interesting discussion points on mechanistic model parameterization and uncertainties (e.g., the role of the curvature for electron transport, the Jc,max/Vc,max ratio, photosynthesis acclimation, forest stand heterogeneity). It is clearly shown that translating leaf-level responses of CO2 effects to the ecosystem scale is very misleading and most important the study provides mechanistic explanations for the differences. The search for the reasons and the clear explanations provided concerning subcomponents of the photosynthesis model (e.g., Rubisco vs. electron transport limited, or acclimation of photosynthetic capacity) represents an innovative approach, which I did not see before in the literature. For these reasons, beyond the importance of estimating GPP in ambient and eCO2 conditions that will serve future studies in the context of the EucFACE experiment, the article represents an important piece of work for the mechanistic understanding of ecosystem responses to elevated CO2.

The article is overall very well written and presented. In summary, I think the manuscript is making an important contribution to the field and I sincerely congratulate the authors for this nice piece of work. In the following, I just have a number of minor comments that can be helpful to improve further the presentation of this work.

Sincerely,

Simone Fatichi

*Response: We would like to thank the reviewer for his detailed and positive evaluation of our work. We have modified the manuscript according to his comments.*

Minor comments

P.2 Line 34-37. The difference between canopy scale "direct response" of +11% and the mean actual response of 6%, while very clear in the manuscript, it is not so clear at the abstract level. Maybe introducing the concept of "uncertainty" associated with the variability across rings or something associated to the "actual field response" according to the experimental configuration may help.

*Response: We have changed the text to read:*

*'After taking in account the baseline variability in leaf area index across plots, we estimated a field GPP response to $eC_a$ of 6% with a 95% confidence interval (-2, 14%).'*

P.3 Line 51. The "hence" here is out of place, because the causality is not straightforward. An increase in carbon uptake does not necessarily lead to an increase in the amount of carbon stored in the ecosystem. The authors are well aware of this. Something like "which in turn could potentially increase. . ." will be more correct.

*Response: We have changed this line to read:*

*"These physiological responses at the leaf scale can increase ecosystem carbon uptake, which in turn may result in increased carbon storage in the ecosystem, mitigating against the rise in $C_a$."*

P.3. Line 57. A short overview of main disagreements between various studies is provided in Fatichi et al. 2019.

*Response: We thank the reviewer for bringing the paper to our attention. The citation is now added to the paper.*

P.3. Line 57-68. I think this paragraph would benefit from referring to the estimates of global terrestrial C sink. While the attribution of the land C-sink is still debated, an average C-sink of 20–30 g C year-1 m-2 over vegetated land in the last decades is not a detail in the overall story about eCO2.

*Response: We have referred to the land carbon sink in the text:*

*'Similarly, the global carbon budget indicates a strong sink for carbon on land (Le Quéré et al., 2018).'*

P. 4. Line 80. While, practically, I would agree in defining the response of GPP to eCO2 an upper bound. Theoretically, this is not a limit, if for some reason, plants in eCO2 conditions will be able to do maintenance with half of the respiration costs, then the NPP response could be larger than the GPP response. I think a "reference value" is more correct than an "upper-bound".

*Response: We have modified the line to read:*

*"The response of GPP is important because it provides a reference point against which to compare the response of other components of ecosystem carbon balance, such as above-ground growth."*

P.4. Line 115 and P.6 Line 161 and 166. Yang et al. 2019 is missing from the reference list, overall, I would avoid referring to papers, which are not published.

*Response: The paper is now accepted at Tree Physiology (https://doi.org/10.1093/treephys/tpz103). We have added the reference to the list.*

P.4 Line 116 and P.6 Line 170-171. I would not mix the "meteorological forcing" with the "model parameterization". The two aspects are different from a modeling perspective, one represents the inputs to the model, the other (e.g., physiological and structural attributes) represents model parameters, or prognostic variables if these are time dynamics and computed in the model. One can use the same model parameterization with different meteorological inputs and the other way around.

*We agree with the reviewer and deleted 'meteorological' in the sentence.*

P.6 L.173. Figure 2b. I am strongly encouraging to avoid using a linear interpolation for soil moisture values at least in its graphical representation. Soil moisture temporal dynamics have a fast and strong response to rainfall events. Linearly interpolating biweekly value is creating a misleading perception of the real temporal dynamics of soil moisture. I would prefer to have just the points when the soil moisture values have been collected rather than the current representation where raising and descending soil moisture dynamics are often unrealistic.

*Response: We have adapted the suggestion by plotting the soil wwater content as dots and revised the figure legend. Thank you for this suggestion. We experimented with removing the lines from this plot as suggested but it is hard to see anything without the lines.*

[Figure]

P.7. Line 220. Just a suggestion. Maybe the Fig. S1 could be included in the main manuscript.

*Response: We are glad to see that the reviewer likes fig s1. The fig works well as a conceptual figure showing how the model works but does not related directly to the inputs and results of the model. We thus decide to leave it in the supplementary material.*

P.9 Line Table 1. I know that in literature it is quite typical to report μmol only. However, this is not very precise, especially when we are dealing with photosynthesis. I would suggest to explicitly say μmol of what, e.g., μmol-CO2, μmol-H20, μmol-electrons or better μmol-Eq. as in the original Farquhar et al. 1980.

*We have revised the table as suggested:*

*Table 1. Summary table of parameter definitions, units, and sources used in this study.*

| Parameters | Definitions | Units | Values | Eqn. |
|---|---|---|---|---|
| $\alpha_J$ | Quantum yield of electron transport rate | μmol electron μmol$^{-1}$ photon | 0.30 | S7 |
| $a$ | Fitted slope of LA and DBH | m$^2$ m$^{-1}$ | 492.6 | 4 |
| $a_{abs}$ | Absorptance of PAR | fraction | 0.825 | S4 |
| $b$ | Fitted index of LA and DBH | - | 1.8 | 4 |
| $c_D$ | Slope of $V_{cmax}$ to $D$ | kPa$^{-1}$ | 0.14 | 3 |
| $\Delta S$ | Entropy factor | J mol$^{-1}$ K$^{-1}$ | 639.60 ($V_{cmax}$); | S5 |

| | | | | |
|---|---|---|---|---|
| $E_a$ | Activation energy | J mol$^{-1}$ | 638.06 ($J_{max}$) 66386 ($V_{cmax}$); 32292 ($J_{max}$) | S5 |
| $g_{1.max}$ | Maximum $g_1$ value | kPa$^{0.5}$ | 5.0 | 2 |
| $H_d$ | Deactivation energy | J mol$^{-1}$ | 200000 | S5 |
| $\theta_J$ | Convexity of electron transport rate to $Q_{APAR}$ | - | 0.48 | S8 |
| $\theta_{max}$ | Upper limit of soil water content above which $g_1$ is maximum | - | 0.240 | 2 |
| $\theta_{min}$ | Lower limit of soil water content below which $g_1$ is zero | - | 0.106 | 2 |
| $J_{max.25}$ | Value of $J_{max}$ at 25°C | μmol electron m$^{-2}$ s$^{-1}$ | 159 | 3 |
| $k_T$ | Sensitivity of $R_{dark}$ to temperature | °C$^{-1}$ | 0.078 | S6 |
| $q$ | The non-linearity of the $g_1$ dependence of $\theta$ | - | 0.425 | 2 |
| $R_{day.25}$ | Light respiration rate | μmol C m$^{-2}$ s$^{-1}$ | 0.9 | S6 |
| $R_{dark.25}$ | Dark respiration rate | μmol C m$^{-2}$ s$^{-1}$ | 1.3 | S6 |
| $R_{gas}$ | Gas constant | J mol$^{-1}$ K$^{-1}$ | 8.314 | S5 |
| $V_{cmax.25}$ | Value of $V_{cmax}$ at 25°C | μmol C m$^{-2}$ s$^{-1}$ | 91 (ambient); 83 (elevated) | 3 |

P.10. Line 286-290. Did you check if with MAESPA you get the same +33% of leaflevel photosynthesis if you simulated the same environmental conditions of the 600 A-Ci curves? Very likely, yes, because these are used to estimate the photosynthesis parameters, but just as a double check.

*Response: The leaf gas exchange model in MAESPA is the same leaf-scale model as the 'photosyn' function implemented in the 'plantecophys' R package. We did not check the full MAESPA model but checked the leaf gas exchange model in the R package, which can reproduce the 33% value depending on parameterisation.*

P.10 Figure 6 and 7. In Fig.7 is reported incident PAR and in Fig. 6 absorbed PAR, even though one refer to the stand scale and the other to the leaf-level, I think it would have been better to use either absorbed or incident PAR in both of them for comparison.

*Response: We have changed Fig 6 to PAR so that both figures are directly comparable.*

[Figure]

P. 11. Line 318. From the Supp. Material, the curvature for electron transport θj is also used as curvature and for overall photosynthesis (Eq. S8). These two values are typically different in models (e.g., Bonan et al 2011). This needs to be specified in the manuscript as well. The reference θj =0.85 is typically assumed for the curvature and for the overall photosynthesis, rather than for the curvature of electron transport, which is typically lower in some models (0.7, Bonan et al 2011, Fatichi et al 2016). This needs to be discussed.

*Response: We have modified L320-321 to read:*

*"We explored this effect by investigating the effect of varying the convexity, $\theta_J$, which is assumed to be the same as the convexity of overall photosynthesis."*

*and L422-423 to read:*

*"The parameter value we fitted to data measured in situ ($\theta_J$ = 0.48) is lower than the value commonly assumed in the models (e.g., 0.7 in Bonan et al., 2011). Note that some model studies assume that $\theta_J$ to be lower than the convexity of overall photosynthesis (typically over 0.8; e.g., 0.9 in Medlyn et al., 2002; 0.85 in Haverd et al., 2018). Here we assumed that the convexity of electron transport rate and overall photosynthesis are the same (see Supplementary Text S1 for details)."*

*We added justifications in Supplementary Text S1:*

*'The assumptions of the quantum yield and convexity being the same between J and overall photosynthesis are further explored by comparing the photosynthesis predicted by 'photosyn' function with the fitted $\alpha_A$, and $\theta_J$ to*

*the measured light response curve. There's good agreement with a root mean square error of 2.3 μmol m$^{-2}$ s$^{-1}$ and a R$^2$ of 0.92, suggesting the assumptions are appropriate in our site.* '

P. 12. Line 377. I am honestly impressed by the inter-ring differences in GPP. I think these are mostly related to the relative small size of the rings. Or better, the size is quite large in comparison to experimental capabilities but relative small to average forest stand heterogeneities.

*Response: Despite the relatively consistent overstorey vegetation, this mature forest has remained unmanaged for at least over 90 years, subject to native and variable environmental fluctuations. We therefore believe that spatial heterogeneity is the major driver of the inter-ring variability in GPP.*

P.12 Line 388. Renchon et al 2018 is not in the reference list.

*Response: We have added the paper to the reference list.*

Eq (S3) The denominator should be Ci+2Γ rather than Ci+ Γ (e.g., Wang and Leuning, 1998, Dai et. 2004, Bonan et al 2011);

*Response: Thanks, the equation has been corrected.*

**Response to reviewer 2**

This manuscript syntheses a large amount of data from the EucFACE project to examine the effects of Rubisco- versus RuBP limitation on photosynthesis under elevated CO2. The authors present leaf-level measurements, leaf-level modeling, and canopy scale modeling of ambient versus elevated CO2 conditions to illustrate that current projections of GPP under elevated CO2 are overestimated in mature forests due to biases towards light-saturated leaves. This work is scientifically relevant and pedantic. I want to commend the authors on their efforts and have minor suggestions to improve the presentation and make the work clearer to a wider audience.

The introduction is extremely well written and provides appropriate context for the work being conducted within the manuscript. The methodology is thoughtfully presented, and justification was given for parameter choices in the model. The amount of data used to represent the system is commendable and I appreciate the attention to detail. I find the presentation of soil moisture to be the weakest element of the methodology and would recommend a little more attention paid to it as it is one of the few varying parameters between the replicates. The presentation of the results would be strengthened by more clearly delineating measurements vs. leaf scale modeling vs. canopy scale modeling. I would personally be very interested in seeing some of the rawer data forms (e.g., timeseries of canopy model) in addition to the synthesized percent changes. While this may be a question of style, I found the figure captions to contain relevant information that was missing from the text. I would include more of that information in the text for clarity. Figures are adequate, but the figure legends are not descriptive (esp. Fig 2 and 4-7) and the long captions make it difficult to distinguish between the different replicates, responses, etc.

Overall well done and I'm excited to see this work published!

*Response: We appreciate the reviewer's detailed and positive evaluation of our work. We would like to thank the reviewer and have modified the manuscript according to the comments.*

INTRODUCTION
L94 -95 Can you please give an example of the ranges of Jmax:Vmax ratio found in these cited works to show how much it deviates from the normally adopted ratio of 2?
*Response: We have changed the text to read:*
*'However, recent studies have suggested the $J_{max}$:$V_{cmax}$ ratio varies systematically across forest ecosystems and can range from 1 to 3 (Kattge and Knorr, 2007; Ellsworth et al., 2012; Kumarathunge et al., 2018)'*

METHODS
L132: Is the repo unchanged or should the reader be directed to a certain commit version?
*Response: The repo will remain unchanged. Further development of the model will be through other branches of the repo.*

L162: You do not introduce the variable D until line 172 and do not provide units.
*Response: We thank the reviewer for highlighting this, we have introduced D on line 155.*

L164: Can you please clarify the choice between Jmax and Vcmax here in the Vmax,

t parameter?

*Response: There are measurements on Jmax25, Vcmax25, and their temperature dependence. We correct Jmax25 and Vcmax25 based on the leaf temperature to derive Jmax and Vcmax following Eqn S5. These values of Jmax and Vcmax are then reduced by VPD (a bit more explanation….). We added reference to Text S1 in the manuscript.*

L175-177: I would introduce the equipment and measurement heights before the frequency, but this is a minor point.

*Response: We have changed the ordering as suggested.*

L186: In Fig 2 you present that you use 150cm neutron probe measurements that were conducted biweekly and linearly interpolated, but here you say these measurements were not gap-filled? While I do not think this would majorly affect your results, averaging the soil moisture over the entire 150 cm profile seems problematic as you are giving equal weight to regions that are likely to contain significantly less root biomass. Would it not be more fitting to use a weighted average based on the below-ground biomass distribution to represent the soil moisture that the tree actually "feels"? Would this have changed your g1?

Response: We did not account for root distribution. Instead, we tested the g1 - SWC relationship using SWC averaged over different depths and found that 150cm has the best fit. This result is not shown in the paper but is as below.

[Figure]

L202: Minor point of convention – normally see DBH represented in [cm]. I assume you used DBH in [cm] in your allometry in Eq 4?

*Response: That was a typo and we have now fixed the unit to be cm, thanks.*

L214-215: Fit statistics of this allometry?

*Response: We have modified the sentence to read:*

*'The values obtained via fitting for a and b were 492.6 and 1.8 respectively, with a root mean square error of 14.4 $m^2$ and $R^2$ of 0.83'*

L225: No, no to citing an "in prep" when you seem to be presenting this data in this work.

*Response: We replaced the "in prep" citation with Ellsworth et al. (2017) who also use these data. A much more detailed manuscript is in preparation and we had hoped to be able to cite that, but the 2017 citation is also appropriate.*

L278 Misspelling of ntheta_min

*Response: Thanks, this has been fixed.*

L246: Please expand up on the statement "within two weeks without rain" – was there some selection of points that happened based on this? I'm bit confused with the g1 and soil moisture match up.

*Response: We have modified the text to read:*

*'The $g_1$ values were related to the nearest measurements of $\theta$ (within two weeks). In all cases, there has been no rainfall between $g_1$ and $\theta$ measurement dates. '*

L255: Missing commas.

*Response: It was not clear to us what the reviewer is referring to.*

RESULTS

The results presentation is somewhat difficult to follow given the large number of simulations and measurements spanning scales. Figure 4's mix of bar and point measurements is difficult to follow. Separating data measurements from modeled responses could help the reader follow better what is derived from models and what was an actual measured response. I appreciate the color coding between leaf and canopy measurements. Also, making a clearer distinction between the leaf-level models using the R package and the canopy scaled measures made using MAESPA would be helpful.

*Response: We have changed the bottom row in fig4. Now all modelled results are in bars and observations in points. We have further clarified this in the caption. This should also help separate the results from leaf and canopy models.*

*'The bars represent model outputs while points represent observations.'*

[Figure]

'

DISCUSSION

L397-398: I would love to see the timeseries that illustrates this stomatal closure at the canopy scale.
*Response: These data were originally presented by Gimeno et al. (2016), who show a timeseries in their Figure 2. The data themselves are also publicly available so that the reader can make their own plots. See also Yang et al. (2019) who explore the relationship with VPD.*

L405-418: Ah ha! This was the presentation of J:V ratio ranges that I was waiting for. I would still suggest adding in a range to the introduction so that the reader is primed to consider how variability in J:V could impact these modeled responses to eCa.
*Response: We have addressed this comment above.*
*'However, recent studies have suggested the $J_{max}$:$V_{cmax}$ ratio varies systematically across forest ecosystems and can range from 1 to 3 (Kattge and Knorr, 2007; Ellsworth et al., 2012; Kumarathunge et al., 2018)'*

L420: Yes, but would how would the way you averaged your soil moisture values affect this value?
*Response: We think that the reviewer may have confused the convexity parameter $\theta_J$ with soil moisture content $\theta$. Unfortunately, $\theta$ is the most commonly used symbol for the terms in both fields. We edited the definition of the terms in Table 1 to clarify that values of theta, theta max and theta min refer to soil moisture content.*

L449-451 I agree with this statement, but can you be more clear about what "uncertainty" you are referring to? Are you talking about uncertainty in our forcing variables for models; structural uncertainty in the models; both?
*Response: Here we specifically focused on the variability in the measurements (i.e., inter-ring variability in this study). We clarified this further in the text:*
*'Secondly, the inherent ring-to-ring variation in this natural forest stand is even higher than the GPP response, which highlights the importance of considering both the effect size and uncertainty in the observations than to focus on statistical significance.'*

L462-463 Falsify those model simulations? Should we just throw the models in the trash or can we focus on an improvement in the model structure in order to capture these transitions between Rubisco limitation and RuBP regeneration? Or could it be also that there are other structural differences between those models and the explicit canopy structure of MAESPA?
*Response: As the reviewer highlights, we are falsifying model assumptions, not the model as a whole. Thus, we do not advocate trashing the models, but rather we aim to identify ways forward for model improvement. We modified the text to read:*
*'With our results, it is possible to falsify some of the assumptions made in these model simulations and identify directions for model improvement.'*

FIGURES
L708-711 Fig 2: You give no clear indication about what the different line colors mean. I assume these are replicates, but I am not clear about if these are elevated or control plots. You briefly mention ring numbers in text, but the figure would be improved if you make this distinction more visually apparent.

*Response: We apologise for the confusion. These details were inadvertently omitted from the caption. We now added*

*'Each line colour marks a different plot. Red colours show elevated $CO_2$ plots (treatment), while blue colours show ambient $CO_2$ plots (control).'*

L714: "error bars represent standard error of fitted values" I'm a bit confused by this statement.

*Response: The observations were grouped by date and treatment before fitting. Only one g1 was fitted to each group of data (as stated in the method section 23.3). As a result, the fitting has an uncertainty or error. We used standard error from each fitting to quantify the uncertainty. We added the following to the figure caption:*

*"g1 parameter values are fitted to data grouped by month and treatment."*

L717: This figure is especially hard to follow and the mix of bar and points is difficult. I would suggest adding further groups to help identify measurements vs. leaf scale modeling vs. canopy scale modeling.

*Response: We have addressed this in the earlier comment from the reviewer. Now the bars represent model and points represent observations. We further clarified this in the caption.*